**Data Availability Statement:** The analyses presented in this paper are based on healthcare and driving datasets which are available upon approval from the respective Data Stewards. Researchers who meet the criteria for access to confidential data can accesses the data used for

# Syncope and subsequent traffic crash: A responsibility analysis

**John A. Staples** [1,2]*, **Shannon Erdelyi**[3], **Ketki Merchant**[1], **Candace Yip**[1], **Mayesha Khan**[1], **Donald A. Redelmeier** [4,5], **Herbert Chan**[3], **Jeffrey R. Brubacher**[2,3]

1 Department of Medicine, University of British Columbia, Vancouver, Canada, 2 Centre for Clinical Epidemiology & Evaluation, Vancouver, Canada, 3 Department of Emergency Medicine, University of British Columbia, Vancouver, Canada, 4 Sunnybrook Research Institute, Toronto, Canada, 5 Department of Medicine, University of Toronto, Toronto, Canada

* john.staples@ubc.ca

## Abstract

### Background

Physicians are often asked to counsel patients about driving safety after syncope, yet little empirical data guides such advice.

### Methods

We identified a population-based retrospective cohort of 9,507 individuals with a driver license who were discharged from any of six urban emergency departments (EDs) with a diagnosis of 'syncope and collapse'. We examined all police-reported crashes that involved a cohort member as a driver and occurred between 1 January 2010 and 31 December 2016. We categorized crash-involved drivers as 'responsible' or 'non-responsible' for their crash using detailed police-reported crash data and a validated responsibility scoring tool. We then used logistic regression to test the hypothesis that recent syncope was associated with driver responsibility for crash.

### Results

Over the 7-year study interval, cohort members were involved in 475 police-reported crashes: 210 drivers were deemed responsible and 133 drivers were deemed non-responsible for their crash; the 132 drivers deemed to have indeterminate responsibility were excluded from further analysis. An ED visit for syncope occurred in the three months leading up to crash in 11 crash-responsible drivers and in 5 crash-non-responsible drivers, suggesting that recent syncope was not associated with driver responsibility for crash (adjusted odds ratio, 1.31; 95%CI, 0.40–4.74; p = 0.67). However, all drivers with cardiac syncope were deemed responsible, precluding calculation of an odds ratio for this important subgroup.

this study through Population Data BC. More information on data access procedure is available at: https://www.popdata.bc.ca/data_access.

**Funding:** This study was supported by the Canadian Institutes of Health Research (grant number PJT-148849; cihr-irsc.gc.ca). JS' salary was supported by the Vancouver Coastal Health Research Institute (vchri.ca) and by a Health Professional-Investigator Award from Michael Smith Health Research BC (msfhr.org). JB was supported by Michael Smith Health Research BC and the British Columbia Emergency Medicine Network (bcemergencynetwork.ca). The funders had no role in study design, data collection and analysis, decision to publish, or preparation of the manuscript.

**Competing interests:** The authors have declared that no competing interests exist.

## Conclusions

Recent syncope was not significantly associated with driver responsibility for traffic crash. Clinicians and policymakers should consider these results when making fitness-to-drive recommendations after syncope.

## Introduction

Syncope is characterized by a transient loss of consciousness and postural tone, symptoms incompatible with the safe operation of a motor vehicle [1]. One in three people experience syncope over their lifetime and 9% of patients experience another syncope within one year [2, 3]. Driving safety after syncope has long fascinated clinicians, policymakers and the public because the compelling mental image of an incapacitated driver careening into oncoming traffic makes obvious that the risk of injury is borne by all road users (not just the incapacitated driver) [4, 5]. As a result, physicians are often asked to counsel patients about driving safety after syncope and, in some jurisdictions, clinicians must report such events to driver licensing authorities [6, 7].

Most prior studies of syncope and driving are limited by methodological flaws including lack of an appropriate control group, bias from crash self-reporting, insufficient sample size, and limited applicability to patients typically seen in clinical practice [8–10]. Two recently published population-based studies address many of these limitations and together increased the number of patients in the published literature on syncope and driving by 26-fold [11]. A Danish study found that 41,039 individuals visiting a hospital or emergency department with first-episode syncope subsequently sustained crash injuries at nearly twice the rate observed in the general population [12]. In contrast, we found that 9,223 drivers visiting an emergency department in Canada for first-episode syncope had a risk of subsequent crash no different than 34,366 drivers with an emergency visit for a condition other than syncope [13]. These discordant results might be explained if the Canadian patients masked a post-syncope increase in 'crash risk while driving' by reducing subsequent road exposure (the hours or miles driven per week) [14]. Road exposure might be reduced because of restrictions by driver licensing authorities, through physician warnings not to drive, or via driving self-restriction by concerned patients [15–17]. Understanding whether syncope patients are 'as safe as the average driver' or 'low mileage drivers who are dangerous when on the road' is particularly relevant for clinicians and policymakers making fitness-to-drive decisions, yet almost all studies on syncope and crash risk lack data on road exposure.

Responsibility analysis is a well-established method that inherently accounts for road exposure when evaluating risk factors for traffic crash [18–20]. When mitigating factors are present (e.g. icy roads, poor illumination, reckless driving by others) and the driver observed all applicable road laws, the crash is assumed to have occurred for reasons beyond the driver's control and the driver is deemed 'non-responsible' for the crash. When external mitigating factors are absent or when the driver violated road laws, the driver is deemed 'responsible' for the crash. Responsibility analyses automatically account for road exposure because all crash-involved drivers must have been driving at the time of the crash (**Appendix, S1 File**). Exposures present with greater frequency among responsible drivers are believed to increase crash risk (**Fig 1**). Intoxication [19–23], distraction [24], sleep deprivation [25] and other well-established risk factors for crash are associated with crash responsibility, highlighting the utility and face validity of responsibility analyses.

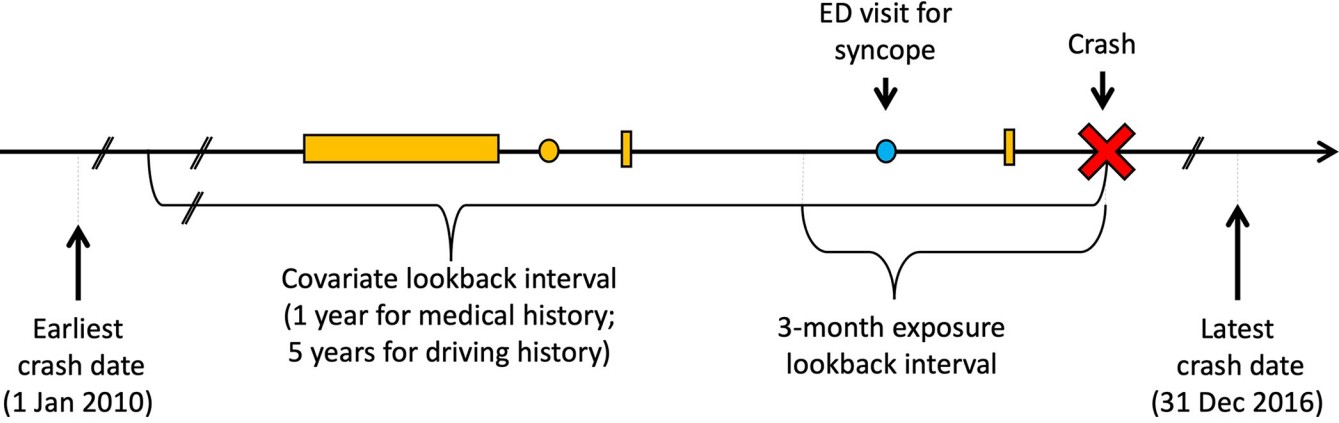

**Fig 1. Study schematic.** Time depicted from left to right. Lookback intervals are anchored on index crash date (red X). The main analysis considered a crash-involved driver 'exposed' if that driver had an ED visit for syncope (blue dot) in the 3 months prior to crash. Comorbidities, medications, and driving history are established using hospitalizations (horizontal orange rectangle), physician visits (orange circle) and contraventions (vertical orange rectangle) in the covariate lookback interval.

We sought to evaluate crash risks while accounting for the possibility that patients reduce their road exposure in the weeks after syncope. Accordingly, we conducted a responsibility analysis evaluating the association between syncope and subsequent motor vehicle crash.

## Methods

### Nested study design

We nested the responsibility analysis within a previously described population-based cohort study of patients with an emergency department visit for first-episode syncope [13]. The syncope cohort included individuals with a driver license who received a discharge diagnosis of 'syncope and collapse' during a visit to one of six urban EDs in British Columbia (BC), Canada, between 1 January 2010 and 31 December 2015. Individuals with age ≤18 years, a full BC driver license for <1 year, a prior emergency visit for syncope between 2007 and 2009, or an index hospitalization length-of-stay >7 days were excluded from the syncope cohort.

All syncope cohort members who were also involved as a driver in a police-attended crash between 1 January 2010 and 31 December 2016 were evaluated as part of the responsibility analysis. We excluded: i) crashes occurring on the date of the syncope ED visit because the temporal sequence of syncope and crash in this situation was often ambiguous; ii) commercial vehicle crashes; and iii) crashes for which we had <1 year of administrative health data prior to crash to identify exposures, comorbidities and other potential confounders. Individuals could contribute to the analysis more than once if they were involved in multiple eligible crashes.

### Study data

We obtained police-reported crash data for all police-attended crashes involving a cohort member as a driver [27]. Police in BC attend all fatal crashes, most serious injury crashes and some crashes involving property damage only. The attending officer completes a structured collision investigation report with detailed information on road type, driving conditions, vehicle condition, crash configuration, the pre-collision action of each vehicle, unlawful or unsafe driving by each crash-involved driver, and the identity of each driver involved in the crash [28].

We linked police-reported crash data to emergency department medical records, administrative health data, and driving history to establish exposure status and to account for potential confounders [13]. Syncope visits for cohort members were identified using administrative health data. Trained abstractors (KM, CY) reviewed all medical records from each cohort member's first syncope ED visit in the study interval to confirm that syncope occurred and to determine the most likely etiology of syncope [13]. We used province-wide individual-level longitudinal administrative health services data to assess comorbidities and prescription medication use at the time of index crash (**S2 File**) [26, 27]. Comorbidities were identified using hospitalizations and physician visits in a 1-year lookback interval (**S3 File**). Prescription medication use was identified using prescription fills dispensed in a 60-day lookback interval [27]. Driving history was established using data from the Insurance Corporation of British Columbia (ICBC), the sole provider of mandatory basic automobile insurance for all vehicles registered in BC and the sole provider of driver licensing services for all drivers licensed in BC. We used these data to establish drivers' licensing details (e.g. license type; issuance, suspension, and expiry dates), prior traffic contraventions (e.g. for alcohol, speeding, or distracted driving), and crash history in a 5-year lookback interval from index crash [26, 27].

## Responsibility scoring

We used a validated scoring tool and police-reported crash data to deem crash-involved drivers 'responsible', 'non-responsible', or of 'indeterminate responsibility' for their crash [21, 27, 28]. The scoring tool assesses 7 factors (road type, driving conditions, vehicle condition, road law obedience, driving task involved, collision type, and contributions from other parties) to determine the degree to which external factors contributed to the crash [28]. Scores for each factor are summed to yield a total score between 7 and 35. Low scores (≤13) indicate few external factors contributed to the crash, and the driver is thus deemed 'responsible' for the crash. Drivers scoring between 13.1 and 15.9 are deemed to have 'indeterminate responsibility' and are removed from the analysis. High scores (≥16) indicate that external factors contributed substantially to the crash and the driver is therefore deemed 'non-responsible' for the crash. Crash responsibility is established independent of any criminal, civil or insurance-based determination of fault. Non-responsible drivers are assumed to be 'randomly selected' by events beyond their control (i.e. they are not the cause of the crash); they are believed to represent drivers who are on the road at the same time but are not involved in the crash [29]. Responsibility analyses thus generate risk estimates similar to standard roadside case-control studies that compare crash-involved drivers to randomly selected, non-crash-involved, 'average' drivers [20, 30].

## Analysis

As in a typical case-control study, we used logistic regression to evaluate the association between crash responsibility (outcome; 'responsible' = 1, 'non-responsible' = 0) and prior ED visit for syncope (exposure; ED visit for syncope present = 1, no visit for syncope = 0; **Fig 1**). For our primary analysis, we selected a 3-month exposure lookback period because most post-syncope driving restriction are between 1 week and 3 months in duration. We adjusted for potential confounders present at the time of crash: Driver sex, age group, and residential neighbourhood income quintile; history of cardiovascular disease, cardiac pacemaker implantation, cardioverter-defibrillator implantation, diabetes, alcohol misuse, or other substance misuse; Charlson Comorbidity Index ≥2; number of physician visits and overnight hospital admissions in the prior year; number of distinct prescription medications and any prescription fills for benzodiazepines or for opioids; full driver license versus a learner or novice license at

the time of crash; documented breath alcohol positivity or police suspicion of alcohol or drug impairment at the time of crash; season in which the crash occurred; the annual proportion of all crash-involved drivers in BC deemed responsible in crash year; and the proportion of days insured, prior contraventions, and crash history of the driver in a 5-year lookback (S4 File).

Subgroup analyses focused on groups relevant to clinicians providing driving advice after syncope. Sensitivity analyses evaluated alternate study intervals, exposure lookback intervals, and responsibility score categorizations. We used R version 4.0, two-sided tests, and p-values <0.05 to establish statistical significance.

### Ethics

The University of British Columbia Clinical Research Ethics Board approved the study and waived the requirement for individual consent (H16-02043). Data were de-identified before release to investigators. It was not appropriate or possible to involve patients or the public in the design, or conduct, or reporting, or dissemination plans of our research. All inferences, opinions, and conclusions drawn are those of the authors and do not reflect the opinions or policies of the Data Stewards.

## Results

The syncope cohort included 9,507 drivers involved in 475 police-reported crashes over the 7-year study interval (Fig 2). Crash-involved drivers had a median age of 52 years and typically possessed a full driver license; the majority held an active vehicle insurance policy, received at least one contravention, and were involved in at least one crash in the 5 years prior to the index crash (Table 1). Among crash-involved drivers, 210 were deemed 'responsible' and 133 were deemed 'non-responsible' for their crash; 132 drivers deemed to have indeterminate responsibility were excluded from the main analysis.

Differences between responsible and non-responsible drivers supported the validity and intuitive appeal of responsibility analysis (Table 1). Established risk factors for crash that do not influence responsibility scores were more common among crash-responsible drivers than among non-responsible drivers (e.g., male sex, a history of prior traffic contraventions, alcohol as a contributing factor, distraction/inattention as a contributing factor; S5 & S6 Files). The proportion of crash-involved drivers deemed responsible was similar to that within the largest prior responsibility study (44% versus 46%, respectively), suggesting missing data on 'contributions from other parties' had a limited effect on overall responsibility scores [27].

An ED visit for syncope occurred in the three months prior to crash for 11 of the 210 crash-responsible drivers (5.2%) and for 5 of the 133 crash-non-responsible drivers (3.8%), suggesting syncope was not associated with subsequent crash responsibility (adjusted odds ratio, 1.31; 95%CI, 0.40–4.74; p = 0.67). Results of subgroup analyses were generally consistent with the results of the main analysis, with no evidence of an association between syncope and crash responsibility even among subgroups likely to be at higher risk of adverse events after syncope (i.e. age ≥56 years, Canadian syncope risk score ≥1; Table 2; S7 File). However, among drivers with a syncope ED visit in the three months prior to crash, all individuals with cardiac syncope and all individuals hospitalized directly from the syncope ED visit were deemed responsible for their crash, precluding calculation of odds ratios (Table 2; S8 File). It remains possible that these specific types of syncope are associated with increased crash responsibility. Our overall results were robust on sensitivity analyses that examined different exposure lookback intervals, responsibility score categorizations and study intervals (Table 3; Fig 3).

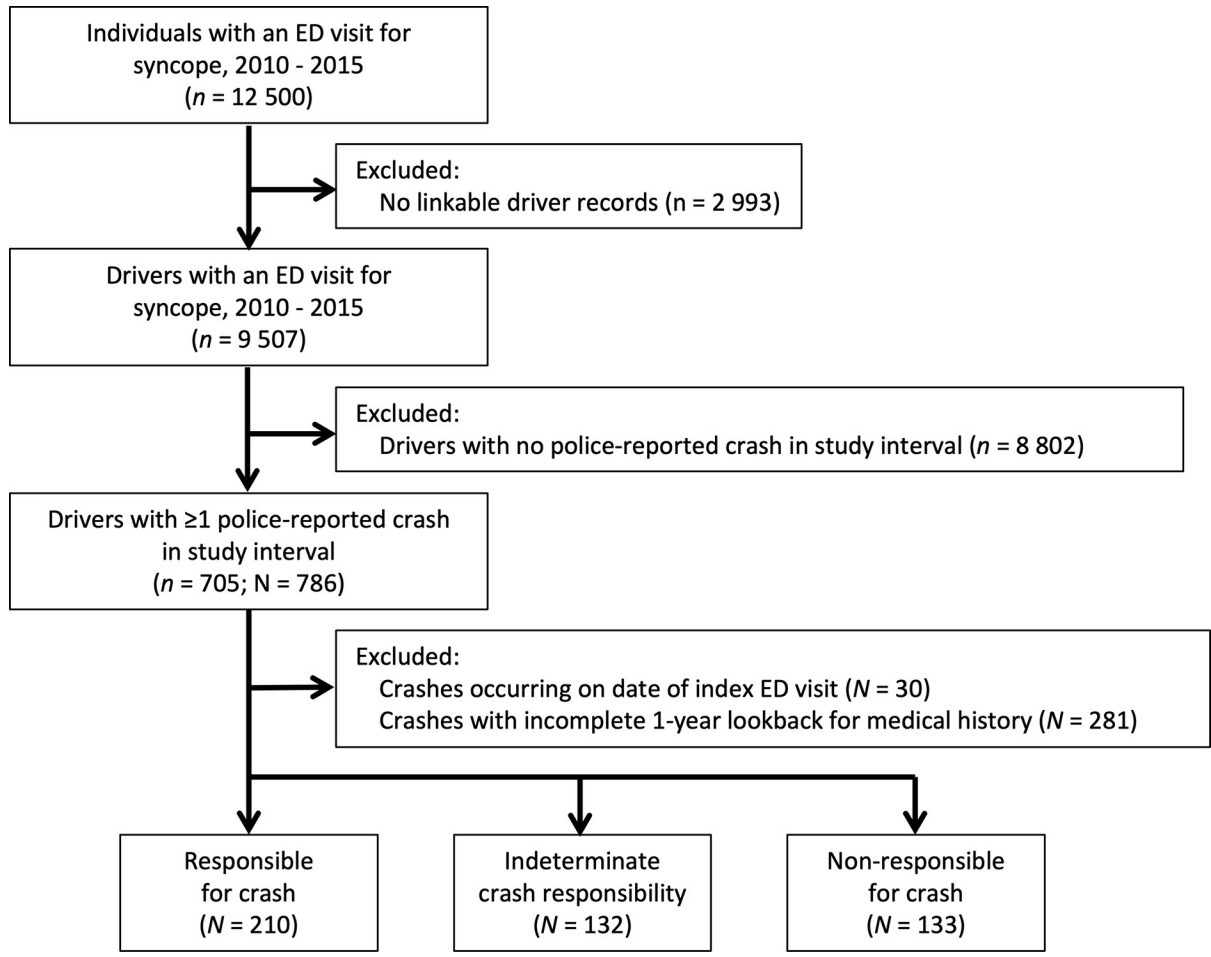

**Fig 2. Study flow diagram.** Drivers were included in analyses more than once if they were involved in multiple police-reported crashes over the study interval. No two drivers in the syncope cohort were involved in the same crash. *n* denotes unique drivers; *N* denotes unique driver-crash combinations; ED = emergency department.

## Discussion

Within a population-based cohort of 9,507 individuals with an ED visit for syncope, we identified 475 police-reported crashes over a 7-year study interval. We found that an ED visit for syncope was not associated with subsequent crash responsibility, suggesting that individuals are not more likely to cause a subsequent traffic collision in the months following an episode of syncope. Subgroup and sensitivity analyses supported the conclusions of the main analysis. However, our results were somewhat imprecise because crashes are rare, suggesting more research is required before patients, clinicians and policymakers can be confident that syncope is not associated with clinically meaningful increases in crash responsibility.

Our findings advance the scientific understanding of syncope and crash risk by complementing and enhancing the interpretation of prior studies [11]. Using the syncope cohort described above, we recently reported that 9,223 individuals visiting the ED for syncope had a crash-free survival no different than 34,366 controls visiting the ED for other conditions (0–30 days, hazard ratio 0.93, 95%CI 0.78–1.11; 30–90 days, hazard ratio 1.07, 95%CI 0.84–1.36) [13]. Using a similar cohort design, Numé and colleagues reported that 41,039 Danish patients visiting the ED or hospital for syncope had a modest increase in the rate of traffic injuries

**Table 1. Characteristics of crash-involved drivers.**

| Characteristic | Drivers deemed responsible for crash n = 210 | Drivers deemed non-responsible for crash n = 133 | Drivers with indeterminate responsibility n = 132 | p-value (responsible vs. non-responsible) |
|---|---|---|---|---|
| **Demographics** | | | | |
| **Median age** (y) [Q1, Q3] | 52 [32, 73] | 49 [32, 65] | 53 [34, 70] | 0.10 |
| **Male sex** | 132 (62.9%) | 67 (50.4%) | 69 (52.3%) | 0.03 |
| **Neighborhood income quintile** | | | | 0.70 |
| First (poorest) | 36 (17.1%) | 30 (22.6%) | 22 (16.7%) | |
| Second | 42 (20.0%) | 29 (21.8%) | 28 (21.2%) | |
| Third | 38 (18.1%) | 23 (17.3%) | 19 (14.4%) | |
| Fourth | 41 (19.5%) | 22 (16.5%) | 26 (19.7%) | |
| Fifth (wealthiest) | 53 (25.2%) | 29 (21.8%) | 36 (27.3%) | |
| **Rural residence** | 33 (15.7%) | 18 (13.5%) | 18 (13.6%) | 0.69 |
| **Medical history** | | | | |
| **≥1 hospitalization in prior year** | 28 (13.3%) | 16 (12.0%) | 15 (11.4%) | 0.85 |
| **≥7 physician visit in prior year** | 139 (66.2%) | 92 (69.2%) | 82 (62.1%) | 0.65 |
| **Charlson co-morbidity score ≥2** | 17 (8.1%) | 14 (10.5%) | 8 (6.1%) | 0.57 |
| **Comorbidities** | | | | |
| Hypertension | 43 (20.5%) | 25 (18.8%) | 23 (17.4%) | 0.81 |
| Cardiovascular disease | 25 (11.9%) | 12 (9.0%) | 14 (10.6%) | 0.51 |
| Cardiac arrhythmia | 15 (7.1%) | 7 (5.3%) | 8 (6.1%) | 0.64 |
| COPD | 12 (5.7%) | 12 (9.0%) | < 5 | 0.34 |
| Diabetes | 8 (3.8%) | 7 (5.3%) | < 5 | 0.71 |
| Cancer | 5 (2.4%) | 5 (3.8%) | 5 (3.8%) | 0.68 |
| Alcohol misuse | 11 (5.2%) | < 5 | < 5 | 0.28 |
| Other substance misuse | 10 (4.8%) | < 5 | 0 | 0.08 |
| **Number of medications** | | | | 0.06 |
| 0 or 1 | 126 (60.0%) | 94 (44.8%) | 97 (46.2%) | |
| ≥2 | 84 (40.0%) | 39 (29.3%) | 35 (26.5%) | |
| **Selected medications** | | | | |
| Benzodiazepines | 16 (7.6%) | 10 (7.5%) | 6 (4.5%) | 1.00 |
| Opioids | 18 (8.6%) | 8 (6.0%) | 9 (6.8%) | 0.51 |
| **Driving history** | | | | |
| **License type** | | | | 0.54 |
| Learner | < 5 | < 5 | 0 | |
| Novice | 31 (14.8%) | 15 (11.3%) | 19 (14.4%) | |
| Full | 176 (83.8%) | 117 (88.0%) | 113 (85.6%) | |
| **Driver experience** (y) [Q1, Q3] | 20 [8, 39] | 18 [7, 35] | 21 [9, 38] | 0.37 |
| **Active license in prior 5y** | 210 (100.0%) | 133 (100.0%) | 132 (100.0%) | - |
| **Insurance policy in prior 5y** | 178 (84.8%) | 115 (86.5%) | 111 (84.1%) | 0.78 |
| **≥1 crash in prior 5y** | 130 (61.9%) | 89 (66.9%) | 77 (58.3%) | 0.41 |
| **≥1 contravention in prior 5y** | 143 (68.1%) | 69 (51.9%) | 74 (56.1%) | 0.004 |
| Alcohol-related | 19 (9.0%) | 8 (6.0%) | 8 (6.1%) | 0.42 |
| Speed-related | 69 (32.9%) | 26 (19.5%) | 37 (28.0%) | 0.01 |

*(Continued)*

**Table 1.** (Continued)

| Characteristic | Drivers deemed responsible for crash n = 210 | Drivers deemed non-responsible for crash n = 133 | Drivers with indeterminate responsibility n = 132 | p-value (responsible vs. non-responsible) |
|---|---|---|---|---|
| Distraction-related | 11 (5.2%) | < 5 | < 5 | 0.48 |

Socioeconomic status and rurality based on the first three digits of residential postal code. Comorbidities considered present if ≥1 hospitalization or ≥2 MSP visits within a 1-year covariate lookback period. Medications considered present if prescription period (defined by medication dispensation date and days dispensed) overlapped with the 60-day period prior to index crash. Driver experience defined as the median years since granted full driver license.

relative to the general population (≤1 month, rate ratio 1.25, 95%CI 0.94–1.67; 1–3 months, rate ratio 1.56, 95%CI 1.30–1.87) [12]. An older case-control study in Washington State evaluated 234 crash-injured drivers aged ≥65 years and 446 matched uninjured controls, finding that driver injury was not associated with antecedent syncope in a 3-year lookback interval (odds ratio 1.8, 95%CI 0.7–5.0) [31]. The current study is unique among this group in that it uses a design which accounts for road exposure; that it produced similar effect estimates suggests there is minimal bias from unmeasured changes in road exposure in this population.

However, our results contrast with the only other published study that compared crash responsibility among syncope patients and controls. Using crash responsibility assigned by the investigating police officer, investigators examined a cohort of 7,750 drivers hospitalized in Maryland between 1994 and 1996 for crash injury and found that crash responsibility was strongly associated with a history of syncope (odds ratio, 4.06; 95%CI, 2.36–7.63); the point estimate somewhat implausibly exceeded that reported for 'alcohol dependence syndrome' (odds ratio, 2.63; 95%CI, 2.01–3.49) [32]. Limitations of that study include uncertain validity of officer-assigned crash responsibility, lack of confirmation of the diagnosis of syncope, uncertain timing between syncope and crash, adjustment only for driver age, issues related to multiple hypothesis testing, and incomplete reporting of study results. The results of our responsibility analysis might reassure decision-makers that syncope patients are less likely to crash than the Maryland study suggests. Other studies on syncope and crash risk either fail to include a control group or simply compare to publicly reported crash risks in the general population [11, 14].

Our findings have implications for clinicians and policymakers charged with making recommendations about fitness-to-drive after syncope. Fitness-to-drive decisions typically depend on the magnitude, not merely the presence, of increased risk: In many jurisdictions, non-zero blood alcohol concentrations (BACs) that double crash risk are not subject to any penalty (i.e. BAC <0.05%); BACs that more than quadruple crash risk only receive a fine or license suspension (i.e. BAC 0.05–0.79%); and only BACs that increase crash risk by more than 7-fold are subject to criminal charges (i.e. BAC >0.08%) [33–35]. The 1.3-fold increase in crash responsibility we observed does not justify new driving restrictions after a syncope ED visit. However, we were unable to exclude a 4-fold increase in risk, suggesting contemporary restrictions for individuals at the highest risk of syncope recurrence while driving are prudent as further research is performed.

The use of responsibility analysis is a unique strength of our study that allowed us to account for road exposure while avoiding selection biases that often characterise recruitment of crash-free control drivers [30]. Our study has many other strengths: We focused on crashes occurring within a population-based cohort; we established crash responsibility objectively using police crash reports and a validated scoring tool; we confirmed the presence and etiology of syncope by performing a structured medical record abstraction of individuals' first ED visit

**Table 2. Subgroup analyses.**

| Subgroup | Proportion with a recent ED visit for syncope among drivers deemed responsible for crash, % (counts) | Proportion with a recent ED visit for syncope among drivers deemed non-responsible for crash, % (counts) | Adjusted odds ratio (95% CI) | p-value |
|---|---|---|---|---|
| **Subgroups defined by driver characteristics** | | | | |
| **Age (years)** | | | | |
| ≤35 | 4.8% (<5) | 2.2% (<5) | 2.09 (0.08, 85.22) | 0.66 |
| 36–55 | 5.1% (<5) | 5.6% (<5) | 2.83 (0.22, 54.24) | 0.44 |
| ≥56 | 5.7% (5/88) | 3.8% (<5) | 0.82 (0.11, 8.55) | 0.85 |
| **Sex** | | | | |
| Male | 5.3% (7/132) | 4.5% (<5) | 1.37 (0.27, 8.00) | 0.71 |
| Female | 5.1% (<5) | 3.0% (<5) | 0.83 (0.09, 8.81) | 0.87 |
| **Cardiovascular disease** | | | | |
| Present | 8.0% (<5) | 0.0% (0/12) | * | * |
| Absent | 4.9% (9/185) | 4.1% (5/121) | 1.20 (0.34, 4.58) | 0.78 |
| **Subgroups defined by syncope ED visit characteristics** | | | | |
| **Diagnostic confidence** | | | | |
| Syncope definite or likely | 3.9% (8/207) | 3.8% (5/133) | 0.92 (0.25, 3.55) | 0.90 |
| Syncope possible or absent | 1.5% (<5) | 0.0% (0/128) | * | * |
| **Syncope subtype** | | | | |
| Vasovagal | 3.4% (7/206) | 2.3% (<5) | 1.55 (0.37, 8.13) | 0.56 |
| Orthostatic | 0.5% (<5) | 1.5% (<5) | 0.11 (0.00, 2.62) | 0.25 |
| Cardiac | 1.0% (<5) | 0.0% (0/128) | * | * |
| Other | 0.5% (<5) | 0.0% (0/128) | * | * |
| **Canadian syncope risk score** | | | | |
| Positive (score ≥ 1) | 1.0% (<5) | 0.8% (<5) | 1.13 (0.07, 30.26) | 0.93 |
| Negative (score = 0) | 4.3% (9/208) | 3.0% (<5) | 1.36 (0.37, 5.67) | 0.65 |
| **San Francisco syncope rule** | | | | |
| Positive (score ≥ 1) | 2.9% (6/205) | 3.0% (<5) | 0.85 (0.20, 3.80) | 0.82 |
| Negative (score = 0) | 2.5% (5/204) | 0.8% (<5) | 3.68 (0.44, 79.42) | 0.28 |
| **Hospitalized on index ED visit** | | | | |
| Yes | 0.5% (<5) | 0.0% (0/128) | * | * |
| No | 4.8% (10/209) | 3.8% (5/133) | 1.06 (0.32, 3.91) | 0.93 |

As for the main analysis, drivers were considered exposed if they had an emergency visit for syncope in the 3 months prior to crash. Subgroups were defined by: a) driver characteristics at the time of crash, or b) comparing mutually exclusive exposure groups (e.g. ED visit for syncope in which abstractors deemed syncope 'definite or likely') to the non-exposed referent group used in the primary analysis (e.g. no ED visit for syncope).

* odds ratios could not be estimated because all exposed drivers in the strata were deemed responsible for their crash.

for syncope; and we had detailed data that allowed us to account for baseline comorbidities, prescription medication use and driver history. Our study also has limitations. First, our findings only apply to individuals who continue to drive after an ED visit for syncope and not to individuals who subsequently ceased driving by choice, following physician advice, or by legal obligation. However, driving cessation after first-episode syncope is rare because the most common types of syncope such as vasovagal or orthostatic syncope are typically not subject to

**Table 3. Sensitivity analyses.**

| Analysis | Adjusted odds ratio (95% CI) | p-value |
|---|---|---|
| **Alternate exposure lookback intervals (months prior to crash)** | | |
| 0–1 month | 1.80 (0.14, 45.37) | 0.66 |
| 0–3 months (main analysis) | 1.31 (0.40, 4.74) | 0.67 |
| 0–6 months | 1.19 (0.58, 2.51) | 0.64 |
| 0–9 months | 0.95 (0.50, 1.86) | 0.89 |
| 0–12 months | 0.97 (0.53, 1.77) | 0.91 |
| **Alternate categorization of responsibility score** | | |
| $\leq$12 versus $\geq$17 | 0.65 (0.15, 3.19) | 0.57 |
| $\leq$13 versus $\geq$16 (main analysis) | 1.31 (0.40, 4.74) | 0.67 |
| $\leq$14 versus $\geq$15 | 1.49 (0.51, 4.71) | 0.48 |
| **Alternate crash eligibility** | | |
| 2010–2016 (main analysis) | 1.31 (0.40, 4.74) | 0.67 |
| 2010–2015 | 1.34 (0.41, 4.88) | 0.64 |

Longer exposure lookback intervals increased the proportion of crash-involved drivers 'exposed' to a prior ED visit for syncope and yielded narrower confidence intervals, but no interval suggested there was an association between syncope and crash responsibility. Similarly, our results were robust to changes in the responsibility score cut-offs used to define crash-responsible drivers and to alternate criteria used to define eligible crashes.

any driving restriction; our results thus apply to most syncope patients seen in the emergency department. Second, our findings are subject to the unverifiable 'randomness assumption' of responsibility analysis which posits that non-responsible crashes are random events caused by external factors [36]. Nevertheless, prior studies suggest that responsibility analyses appropriately identify risk factors for crash [18–20, 23–27, 29]. Third, we lacked data on the

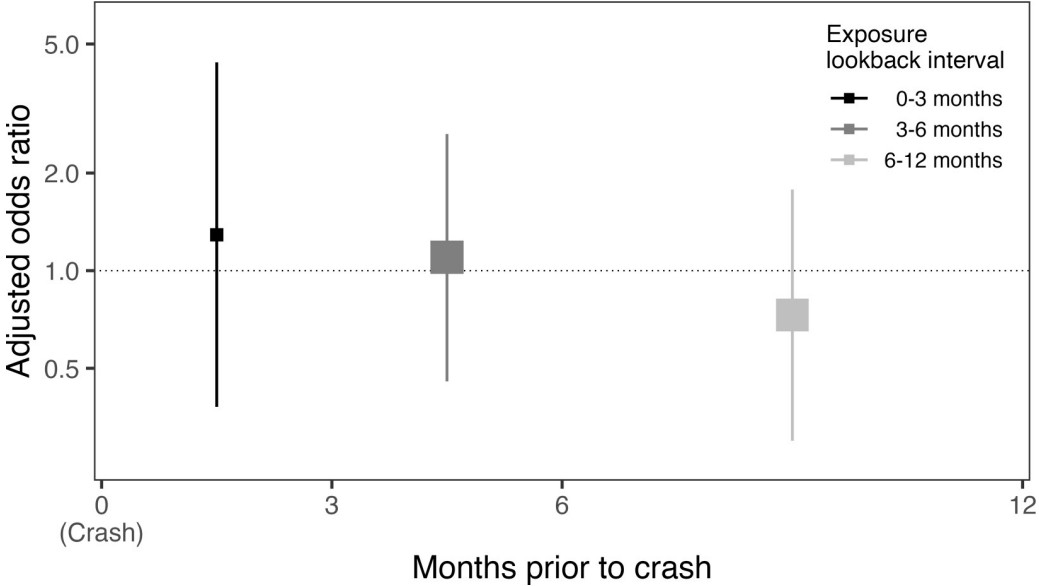

**Fig 3. Forest plot of alternate exposure lookback intervals.** Forest plot of piecewise exposure lookback intervals (from left to right: 0–3 months; 3–6 months; 6–12 months). X-axis depicts the exposure lookback interval; Y-axis, the odds ratio for the association between syncope and crash responsibility; squares, the adjusted odds ratio point estimate (with size reflecting the inverse of the standard error); vertical lines, the 95% confidence interval.

'contribution from other drivers', biasing responsibility scores downward and biasing effect estimates toward the null. Reassuringly, the proportion of our crash cohort deemed responsible aligns with prior studies, suggesting the missing data had minimal effect on our results. Fourth, we had incomplete data on alcohol and non-prescription drug use. We partially accounted for this by adjusting for police-reported alcohol or drug impairment at the time of crash, and by adjusting for medical visits for substance misuse in the year prior to crash. Fifth, we were unable to calculate odds ratios in some higher risk syncope subgroups (e.g. cardiac syncope, hospitalized at time of ED visit for syncope), and it remains possible that these or other subgroups of syncope patients are more likely to cause a subsequent crash. Despite these limitations, our study represents a substantial contribution to the limited available evidence that informs fitness-to-drive decisions after syncope [37].

Medical driving restrictions can substantially decrease patients' quality-of-life [38]. Our findings suggest that individuals with a recent syncope ED visit are no more likely to be responsible for serious motor vehicle crashes than are comparable patients with no recent syncope ED visit, and that expanded driving restrictions after syncope are not warranted.

## Supporting information

**S1 File. Responsibility analysis accounts for changes in road exposure.** Table using hypothetical data to illustrate that responsibility analyses inherently account for road exposure because all crash-involved drivers were driving at the time of the crash. Grey cell highlights the change in inputs as the reader moves from left to right. Odds ratio = (C/D)÷(E/F). Scenario 1 depicts a responsibility analysis of all crashes in a population of 100,000 individuals, 1% of whom have a disease ('exposed'), with identical road exposure between exposed and control individuals. Scenario 2 depicts a responsibility analysis of all crashes among exposed and controls, but with a yearly travel distance among exposed that is half that of controls. The odds ratios produced in Scenarios 1 and 2 are identical and reflect only the relative proportion of crashes for which each group is responsible; they are unaffected by changes in cohort size and road exposure provided the probability of responsibility is not influenced by road exposure. (DOCX)

**S2 File. Data sources.**
(DOCX)

**S3 File. ICD diagnostic codes used to define baseline comorbidities.** We considered comorbidities present if identified in diagnostic coding from ≥1 hospitalization or ≥2 physician visits in a 1-year lookback interval. ICD = The World Health Organization's International Statistical Classification of Diseases and Related Health Problems; ICD9 = ICD, 9th Revision, Clinical Modification (ICD-9-CM) codes; ICD10 = ICD, 10th Revision, Canada (ICD-10-CA) codes; HIV = human immunodeficiency virus; AICD = automated internal cardioverter-defibrillator, CVD = cardiovascular disease.
(DOCX)

**S4 File. Variable definitions.** * indicates the referent category used in regression analyses. We used a logistic regression model to evaluate the association between crash responsibility and prior ED visit for syncope. The equation took the form

$$logit[Pr(Y = 1 | X = x)] = \beta_0 + \beta_s x_s + \sum_{j=1}^{k} \beta_j x_j$$

where
$Y$ = Driver responsibility for crash (1 = responsible, 0 = non-responsible)

$\beta_0$ = Intercept

$\beta_s$ = Regression coefficient that represents the change in the log odds of crash responsibility for drivers with versus without an emergency department visit for 'syncope and collapse' in the 3 months prior to the index crash

$x_s$ = Indicator variable for whether the driver had an emergency department visit for 'syncope and collapse' in the 3 months prior to the index crash (1 = present, 0 = absent)

$\beta_j$ = Regression coefficient representing the change in the log odds of crash responsibility for a 1-unit increase in the j$^{th}$ potential confounder

$x_j$ = Variable representing the value of the j$^{th}$ potential confounder

$k$ = Number of potential confounders.

(DOCX)

**S5 File. Responsibility scores.** Mirrored bar chart comparing scores between responsible and non-responsible drivers for each of the seven components of the responsibility score.
(TIF)

**S6 File. Crash characteristics among responsible and non-responsible drivers.**
Table depicting crash characteristics for responsible and non-responsible drivers. These crash characteristics are reported by police on form MV6020 and aggregated within BC's Traffic Accident System dataset. An example of the data collection form can be found online (Traffic Crash Reports & Overlay Forms [Internet]. North Platte (NE): Accreditation Commission for Traffic Accident Reconstruction; 2022. Accessed 14 Nov 2022 at https://actar.org/resources/reports). We had detailed police-reported crash data for all syncope cohort members but lacked crash data for other drivers involved in the crash. As a result, except where the index driver's data directly suggested contribution from others (e.g., pedestrian error, previous traffic crash), the 'contributions from other parties' factor could not account for other drivers' actions. This may have biased responsibility scores downward and effect estimates toward the null. However, the proportion of crash-involved drivers deemed responsible was similar to that within the largest prior responsibility study (44.2% versus 46%, respectively), suggesting missing data on 'contributions from other parties' had a limited effect on our results (Brubacher JR, Chan H, Erdelyi S, Zed PJ, Staples JA, Etminan M. Medications and risk of motor vehicle collision responsibility in British Columbia, Canada: a population-based case-control study. Lancet Public Health. 2021 Jun;6(6):e374-e385.). Not all displayed data are components of the responsibility score tool. As expected, responsible drivers were more likely to be disobeying road laws and were more likely to be involved in crashes that occurred on dry roads, during optimal weather, in full daylight, and involving only a single vehicle (these are components of the responsibility score). Reassuringly, established risk factors for crash that are not part of the responsibility score are also more common among responsible drivers. For example, human contributory factors, which include alcohol, fatigue, and distraction/inattention, were more common among responsible than among non-responsible drivers (54.3% vs 16.5%; unadjusted odds ratio, 6.00; 95%CI, 3.51 to 10.2; p<0.001). Other established risk factors for crash presented in Table 1 that are not a part of the responsibility score were also associated with crash responsibility, including male sex (62.9% of responsible drivers and 50.4% of non-responsible drivers; unadjusted odds ratio, 1.67; 95%CI, 1.07 to 2.59; p = 0.025) and a prior history of any traffic contravention in a 5-year lookback (68.1% vs 51.9%; unadjusted odds ratio, 1.98; 95%CI, 1.27 to 3.10; p = 0.003).
(DOCX)

**S7 File. Forest plot of subgroup analyses.** X-axis depicts the adjusted odds ratio for the association between syncope and crash responsibility; y-axis, the subgroup; square points, the

adjusted odds ratio point estimate (with size reflecting the inverse of the standard error); horizontal lines, the 95% confidence interval, with arrow heads indicating the confidence interval endpoint is beyond the limit of the x-axis. Syncope was not associated with crash responsibility in any subgroup. *Indicates that all drivers in these strata were deemed responsible for their crash, making it impossible to calculate an odds ratio.
(TIF)

**S8 File. Sensitivity analysis of exposure lookback period for selected subgroups.** *Could not be estimated because all exposed drivers were responsible. ** Too few crashes to estimate parameters in adjusted model.
(DOCX)

**S9 File. Regression coefficients for the main analysis of syncope and crash responsibility.** Main effect is bolded and highlighted in blue. OR = odds ratio, CI = confidence interval, ref = reference category, ED = Emergency Department, BC = British Columbia, AICD = Automated internal cardioverter-defibrillator.
(DOCX)

# Acknowledgments

**Disclaimer:** All inferences, opinions, and conclusions drawn are those of the authors and do not reflect the opinions or policies of the Data Stewards.

# Author Contributions

**Conceptualization:** John A. Staples, Shannon Erdelyi, Donald A. Redelmeier, Herbert Chan, Jeffrey R. Brubacher.

**Data curation:** John A. Staples, Shannon Erdelyi, Ketki Merchant, Candace Yip, Herbert Chan, Jeffrey R. Brubacher.

**Formal analysis:** John A. Staples, Shannon Erdelyi.

**Funding acquisition:** John A. Staples, Shannon Erdelyi, Donald A. Redelmeier, Herbert Chan, Jeffrey R. Brubacher.

**Investigation:** John A. Staples, Shannon Erdelyi, Ketki Merchant, Candace Yip, Herbert Chan, Jeffrey R. Brubacher.

**Methodology:** John A. Staples, Shannon Erdelyi, Donald A. Redelmeier, Herbert Chan, Jeffrey R. Brubacher.

**Project administration:** John A. Staples, Mayesha Khan, Herbert Chan.

**Resources:** John A. Staples, Jeffrey R. Brubacher.

**Supervision:** John A. Staples, Ketki Merchant, Jeffrey R. Brubacher.

**Validation:** John A. Staples, Shannon Erdelyi.

**Visualization:** John A. Staples, Shannon Erdelyi.

**Writing – original draft:** John A. Staples, Shannon Erdelyi, Ketki Merchant, Candace Yip, Mayesha Khan, Jeffrey R. Brubacher.

**Writing – review & editing:** John A. Staples, Shannon Erdelyi, Ketki Merchant, Candace Yip, Mayesha Khan, Donald A. Redelmeier, Herbert Chan, Jeffrey R. Brubacher.

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
