## [Decision Letter · Decision Letter 0]

24 Oct 2022

PONE-D-22-23799Syncope and subsequent traffic crash: A responsibility analysisPLOS ONE

Dear Dr. Staples,

Thank you for submitting your manuscript to PLOS ONE. After careful consideration, we feel that it has merit but does not fully meet PLOS ONE’s publication criteria as it currently stands. Therefore, we invite you to submit a revised version of the manuscript that addresses the points raised during the review process.

Please try to address all the reviewers' comments and concerns.

We look forward to receiving your revised manuscript.

Kind regards,

Quan Yuan, Ph.D.

Academic Editor

PLOS ONE

Journal Requirements:

"This study was supported by the Canadian Institutes of Health Research (grant number PJT-148849). JS' salary was supported by the Vancouver Coastal Health Research Institute and by a Health Professional-Investigator Award from Michael Smith Health Research"

"This study was supported by the Canadian Institutes of Health Research (grant number PJT-148849; cihr-irsc.gc.ca). JS' salary was supported by the Vancouver Coastal Health Research Institute (vchri.ca) and by a Health Professional-Investigator Award from Michael Smith Health Research BC (msfhr.org). JB was supported by Michael Smith Health Research BC and the British Columbia Emergency Medicine Network (bcemergencynetwork.ca). The funders had no role in study design, data collection and analysis, decision to publish, or preparation of the manuscript."

Reviewers' comments:

Reviewer's Responses to Questions

**Comments to the Author**

1. Is the manuscript technically sound, and do the data support the conclusions?

Reviewer #1: Yes

Reviewer #2: Yes

2. Has the statistical analysis been performed appropriately and rigorously? 

Reviewer #1: Yes

Reviewer #2: N/A

3. Have the authors made all data underlying the findings in their manuscript fully available?

Reviewer #1: Yes

Reviewer #2: Yes

4. Is the manuscript presented in an intelligible fashion and written in standard English?

Reviewer #1: Yes

Reviewer #2: Yes

5. Review Comments to the Author

Reviewer #1: It is an interesting study. The authors shown us clearly the data, method and results, we can learn a lot of information from the study. Be sure to say something that is insufficient, maybe the authors should be put figures and table near its corresponding descriptions. This maybe the requirement of the Journal, but which real bring inconvenient for readers.

Reviewer #2: This paper focused on the syncope and subsequent traffic crash, based on the collision investigation report to analyze. The topic is interesting, while there are some issues should be modified.

Some suggestions:

1.Authors should provide the samples for the collision investigation report with detailed information.

2. Authors mentioned “used logistic regression to evaluate the association between crash responsibility and prior ED visit for syncope.” Here should provide the formula and results of the logistic regression model.

3. Authors find that “no significant association between an emergency department visit for syncope and driver responsibility for a subsequent motor vehicle crash”, there are many influencing factors of traffic crashes, how to screen the influence of drivers, road and environmental factors needs to be further explored.

6. PLOS authors have the option to publish the peer review history of their article (what does this mean?). If published, this will include your full peer review and any attached files.

Reviewer #1: No

Reviewer #2: No

---

## [Author Response · Author response to Decision Letter 0]

14 Nov 2022

Author Response Letter for “Syncope and subsequent traffic crash: A responsibility analysis” (Manuscript PONE-D-22-23799, PLoS ONE) accompanying the first revision.

Editor and reviewer comments in bold; author responses in italics. 

EDITOR'S COMMENTS:

Editor: Thank you for submitting your manuscript to PLOS ONE ... we invite you to submit a revised version of the manuscript that addresses the points raised during the review process.

AUTHORS' RESPONSE: We thank the editor for inviting us to submit a revision. We have made the suggested changes including formatting to be consistent with PLOS ONE style templates.

Editor: Thank you for stating the following in the Acknowledgments Section of your manuscript: 

"This study was supported by the Canadian Institutes of Health Research (grant number PJT-148849). JS' salary was supported by the Vancouver Coastal Health Research Institute and by a Health Professional-Investigator Award from Michael Smith Health Research"

We note that you have provided funding information that is not currently declared in your Funding Statement. ... let us know how you would like to update your Funding Statement. Currently, your Funding Statement reads as follows: 

"This study was supported by the Canadian Institutes of Health Research (grant number PJT-148849; cihr-irsc.gc.ca). JS' salary was supported by the Vancouver Coastal Health Research Institute (vchri.ca) and by a Health Professional-Investigator Award from Michael Smith Health Research BC (msfhr.org). JB was supported by Michael Smith Health Research BC and the British Columbia Emergency Medicine Network (bcemergencynetwork.ca). The funders had no role in study design, data collection and analysis, decision to publish, or preparation of the manuscript."

AUTHORS' RESPONSE: We have removed all funding-related text from our manuscript. The Funding Statement contains all details that were previously noted in the Acknowledgements section and contains all funding information for this study.

Editor: In your Data Availability statement, you have not specified where the minimal data set underlying the results described in your manuscript can be found. PLOS defines a study's minimal data set as the underlying data used to reach the conclusions drawn in the manuscript and any additional data required to replicate the reported study findings in their entirety. All PLOS journals require that the minimal data set be made fully available. For more information about our data policy, please see http://journals.plos.org/plosone/s/data-availability.

AUTHORS' RESPONSE: The data sets linked for use in this study are owned by the Data Stewards listed in Appendix Item S2. We now describe data access in the manuscript as follows: 

"The analyses presented in this paper are based on laboratory and healthcare utilization datasets which are available upon approval from the respective Data Stewards. Researchers who meet the criteria for access to confidential data can accesses the data used for this study through Population Data BC. More information on data access procedure is available at: https://www.popdata.bc.ca/data_access."

 

Reviewer #1: It is an interesting study. The authors shown us clearly the data, method and results, we can learn a lot of information from the study. Be sure to say something that is insufficient, maybe the authors should be put figures and table near its corresponding descriptions. This maybe the requirement of the Journal, but which real bring inconvenient for readers.

AUTHORS' RESPONSE: We thank Reviewer #1 for their appreciative comments. We happily leave decisions regarding formatting to the Editor and to the PLoS ONE team. 

 

Reviewer #2: This paper focused on the syncope and subsequent traffic crash, based on the collision investigation report to analyze. The topic is interesting, while there are some issues should be modified. Some suggestions:

1.Authors should provide the samples for the collision investigation report with detailed information.

AUTHORS' RESPONSE: We thank Reviewer#2 for finding our study interesting.

We have now indicated in our appendix (p11) that the detailed data we present in Item S6 is all derived from police crash reports. We now state the number of the police crash report form and provide a reference that allows readers to obtain a version of the reporting form:

"These crash characteristics are reported by police on form MV6020 and aggregated within BC's Traffic Accident System dataset. An example of the data collection form can be found online. [Traffic Crash Reports & Overlay Forms [Internet]. North Platte (NE): Accreditation Commission for Traffic Accident Reconstruction; 2022. Accessed 14 Nov 2022 at https://actar.org/resources/reports]" 

2. Authors mentioned “used logistic regression to evaluate the association between crash responsibility and prior ED visit for syncope.” Here should provide the formula and results of the logistic regression model.

AUTHORS' RESPONSE: We now provide the logistic regression formula for interested readers in Supplemental Appendix (Item S4, p6). We now also provide the results of the main logistic regression model in the Supplemental Appendix (Item S9, p16).

3. Authors find that “no significant association between an emergency department visit for syncope and driver responsibility for a subsequent motor vehicle crash”, there are many influencing factors of traffic crashes, how to screen the influence of drivers, road and environmental factors needs to be further explored.

AUTHORS' RESPONSE: We agree that many factors contribute to crash responsibility. Our revised submission now quantifies the influence of important confounders by presenting the regression coefficients for readers to review (Supplemental Appendix, Item S9, p16). This includes many driver characteristics known to influence crash risk. Road and environmental factors that contribute to the likelihood of crash are also presented in detail in our revised submission (Supplemental Appendix, Item S6, p8). We highlight study limitations including incomplete data on additional potential confounders in our Discussion (p18).

---

## [Decision Letter · Decision Letter 1]

13 Dec 2022

Syncope and subsequent traffic crash: A responsibility analysis

PONE-D-22-23799R1

Dear Dr. Staples,

We’re pleased to inform you that your manuscript has been judged scientifically suitable for publication and will be formally accepted for publication once it meets all outstanding technical requirements.

Kind regards,

Quan Yuan, Ph.D.

Academic Editor

PLOS ONE

Additional Editor Comments (optional):

Reviewers' comments:

Reviewer's Responses to Questions

**Comments to the Author**

1. If the authors have adequately addressed your comments raised in a previous round of review and you feel that this manuscript is now acceptable for publication, you may indicate that here to bypass the “Comments to the Author” section, enter your conflict of interest statement in the “Confidential to Editor” section, and submit your "Accept" recommendation.

Reviewer #1: All comments have been addressed

Reviewer #2: All comments have been addressed

2. Is the manuscript technically sound, and do the data support the conclusions?

Reviewer #1: Yes

Reviewer #2: Yes

3. Has the statistical analysis been performed appropriately and rigorously? 

Reviewer #1: Yes

Reviewer #2: Yes

4. Have the authors made all data underlying the findings in their manuscript fully available?

Reviewer #1: Yes

Reviewer #2: Yes

5. Is the manuscript presented in an intelligible fashion and written in standard English?

Reviewer #1: Yes

Reviewer #2: Yes

6. Review Comments to the Author

Reviewer #1: The research is interesting and deserved to be shared with reads.

Now, all my concerns are addressed. Congratulations.

Reviewer #2: According to the detailed responses, the authors have carefully revised the manuscript, I think this paper can be accepted by the Journal.

7. PLOS authors have the option to publish the peer review history of their article (what does this mean?). If published, this will include your full peer review and any attached files.

Reviewer #1: No

Reviewer #2: No

---

## [Editor Report · Acceptance letter]

4 Jan 2023

PONE-D-22-23799R1 

Syncope and subsequent traffic crash: A responsibility analysis 

Dear Dr. Staples:

I'm pleased to inform you that your manuscript has been deemed suitable for publication in PLOS ONE. Congratulations! Your manuscript is now with our production department. 

Kind regards, 

on behalf of

Dr. Quan Yuan 

Academic Editor

PLOS ONE